# Sharing Soil and Building Geophysical Data for Seismic Characterization of Cities Using CLARA WebGIS: A Case Study of Matera (Southern Italy)

Nicola Tragni [1,2], Giuseppe Calamita [1,*], Lorenzo Lastilla [3,4], Valeria Belloni [5], Roberta Ravanelli [5], Michele Lupo [1], Vito Salvia [1] and Maria Rosaria Gallipoli [1]

1    National Research Council of Italy (CNR-IMAA), 85050 Tito Scalo, Italy; nicola.tragni@imaa.cnr.it (N.T.); michel.lupo@libero.it (M.L.); vito.salvia@imaa.cnr.it (V.S.); mariarosaria.gallipoli@imaa.cnr.it (M.R.G.)
2    School of Engineering, University of Basilicata, 85100 Potenza, Italy
3    Department of Computer, Control and Management Engineering Antonio Ruberti (DIAG), Sapienza University of Rome, 00185 Rome, Italy; lorenzo.lastilla@uniroma1.it
4    Sapienza School for Advanced Studies, 00161 Rome, Italy
5    Geodesy and Geomatics Division, DICEA, Sapienza University of Rome, 00184 Rome, Italy; valeria.belloni@uniroma1.it (V.B.); roberta.ravanelli@uniroma1.it (R.R.)
*    Correspondence: giuseppe.calamita@imaa.cnr.it

**Abstract:** In the context of seismic risk, studying the characteristics of urban soils and of the built environment means adopting a holistic vision of the city, taking a step forward compared to the current microzonation approach. Based on this principle, CLARA WebGIS aims to collect, organize, and disseminate the available information on soils and buildings in the urban area of Matera. The geodatabase is populated with (i) 488 downloadable geological, geotechnical, and geophysical surveys; (ii) geological, geomorphological, and seismic homogeneous microzone maps; and (iii) a new Digital Surface Model. The CLARA WebGIS is the first publicly available database that reports for the whole urban area the spatial distribution of the fundamental frequencies for soils and the overlying 4043 buildings, along with probability levels of soil-building resonance. The WebGIS is aimed at a broad range of end users (local government, engineers, geologists, etc.) as a support to the implementation of seismic risk mitigation strategies in terms of urban planning, seismic retrofitting, and management of post-earthquake crises. We recommend that the database be managed by local administrators, who would also have the task of deciding on future developments and continuous updating as new data becomes available.

**Keywords:** seismic risk; WebGIS; seismic resilience; HVNSR; fundamental frequency; soil-building resonance level; DSM

## 1. Introduction

There is a current acceleration of the global urbanization phenomenon: it is estimated that in 2050 about 66% of the world population will reside in cities. In Europe, about 80% of the population lives in urban areas [1]. Italy is characterized by a large number of medium-sized cities (50,000–200,000 inhabitants) and historical centers of inestimable historical and architectural value that are highly exposed to catastrophic events (e.g., earthquakes, landslides, volcanic eruptions, etc.) and extreme climatic events. Therefore, with the increase in urbanization there is a significant increase in the demand for smart technologies for the management of interventions related to the security of the territory in urban areas [1,2]. The Sendai Framework for Disaster Risk Reduction (UN 2015) [3] and The Paris Agreement, 2030 Agenda for Sustainable Development [4] contain the main references and criteria for risk reduction and constitute a general framework under which it is essential to include national strategies on risk knowledge, assessment, and prevention. Examples of good practices in the adoption of smart technologies to improve

environmental sustainability as well as the mobility and safety of citizens are already present in Europe and Italy in particular [5,6]. Here, administrations have developed strategies for the introduction and pervasive diffusion of digital technologies in urban areas (e.g., smart sensors, IoT, cloud computing), transforming cities into open laboratories and stimulating scientific creativity and technological innovation.

The scientific and technological challenges of the 'CLARA' project have, from a Smart Cities perspective, consisted in (i) developing a systemic approach for the characterization of the main physical properties of the urban subsoil and overlying built environment based on the full integration of the most modern, non-invasive, expeditious, and low-cost geophysical technologies [7–10]; (ii) digital archiving of all geological, geotechnical, geophysical, and engineering data for the city of Matera, acquired during the project; (iii) disseminating all the data and results of the project through the active involvement of public administrations (service oriented approach). WebGIS is one of the most widely used technologies for the dissemination of open data, for multiple purposes and in varied contexts, such as tourism, archeology, agriculture, the environment, etc. [11–15]. Aiming at improving global cooperation and communication with other countries, Shi et al. (2009) [11] designed a WebGIS system to relate the genetic classification of the soils of China to the soil taxonomy. Manna et al. (2020) [12] demonstrated how a geospatial decision support system can assist in the planning and management of olive groves and provide operational support to stakeholders.

The use of WebGIS technologies has also been widespread for natural risk assessment and communication [16–19]. With regard to geo-hydrological risk, WebGIS tools have been used for the analysis and/or the management of the risk deriving from floods or landslides [20–22], for slope stability analysis [23], and for online mapping of unstable rock slopes [24]. Salvati et al. (2009) [25] designed and shared a WebGIS to disseminate information on historical landslides and floods in central Italy. Even in the field of seismic risk, in which the presented work is inserted, WebGIS and geodatabase technologies have been used for the development of interactive tools for the definition of seismic hazard scenarios and risk analysis [26], for the assessment of seismic damage in the seaport of Gioia Tauro [27], and for choosing the optimal routes in the case of a seismic event [28]. Other authors have published databases to share acceleration recordings of earthquakes in urban areas of Kalachori (Greece) [29] and the permeability of fault zones and surrounding protolithic rocks in sites around the world [30]. Although some authors have implemented and published software systems to store and visualize subsoil data to be used in seismic microzonation [31], to the best of our knowledge no databases or WebGIS tools relating to soil-building interaction in urban areas have currently been made public.

The soil-building resonance effect is a well-known and extensively studied phenomenon. It can arise during seismic events when the oscillation frequency of a building is very close to that of the foundation soil, causing an increase in damage [32–35]. The soil-building interaction effect for a single/limited number of closed-spaced buildings has been numerically and experimentally studied [36–39], while for the urban scale as a whole only numerical simulation approaches have been proposed [33,40–46]. Recently, Agea-Medina et al. (2020) [47] evaluated the probability of resonance effect in several districts of municipalities of Alicante and Elche, while other authors have produced soil-building resonance level maps based on numerical relations provided by seismic regulations [48,49], and on extensive collections of experimental data [8]. From a legislative point of view, microzonation studies focus exclusively on the seismotectonic, lithostratigraphic, and geotechnical aspects of shallow soil, completely neglecting the presence and role of buildings [8]. Thanks to seismic microzonation studies, it is possible to know the exact areas susceptible to seismic amplification and instabilities; however, there is no information about areas of cities where the soil-building resonance effect could take place during earthquakes.

In this paper, an open tool with the dual function of a geodatabase and disseminative WebGIS, through which it is possible to visualize and download (i) geological, geotechnical, and geophysical data; (ii) the spatial distribution of the main resonance frequency for

urban soils; (iii) the main vibrational frequencies for the 4043 overlying buildings; (iv) the spatial distribution of the soil–building resonance levels for the urban area of Matera, which represents the innovative core is presented. Therefore, for the first time, the urban environment has been seismically characterized as a unicum (urban subsoil and overlying buildings), and all data are fully usable through CLARA WebGIS. In addition, users can also rely on a Digital Surface Model (DSM) of the city of Matera and its surroundings, which was generated with a cross-sensor multi-view approach from a triplet of optical satellite images. DSMs, which incorporate the natural ground surfaces, buildings, vegetation, and other objects higher than the underlying topographic surface [50,51], can serve as valuable input for the characterization of urban structures. In this way, DSMs address the requirement of municipalities for reliable and up-to-date information for land-use and infrastructure planning, the creation and continuation of development plans, and the overall monitoring of changes [52]. The increasing availability of new high-resolution optical spaceborne sensors allows for the creation of precise DSMs (such as the one included in the CLARA WebGIS), ensuring low cost, speed of data acquisition and processing, and relaxed logistic requirements [53]. A detailed presentation of the experimental design and the methodology adopted to produce the soil-building resonance map and to generate the DSM can be found in Gallipoli et al. (2020) [8] and Lastilla et al. (2021) [54], respectively.

## 2. CLARA WebGIS: Data Sources

The CLARA WebGIS, which is accessible at https://smartcities-matera-clara.imaa.cnr. it/, is populated with two types of data sources: (i) open data made available by public institutions and (ii) experimental geophysical data about shallow soils and buildings, collected both in previous geological studies supporting territorial planning and within the project, organized in 25 layers with specific vector geometries (Table 1). The queries to the database can be graphically formulated using the hand cursor icon, avoiding the need to use the SQL language.

**Table 1.** Characteristics of all objects present in the CLARA WebGIS.

| CLARA | Vector Geometry | # | Download |
|---|---|---|---|
| OD Age of construction | point | 2648 | - |
| OD Typology | point | 2648 | - |
| OD State of conservation | point | 2648 | - |
| RSDI Height max | point | 4522 | - |
| RSDI edifici is | polygon | 11,802 | - |
| RSDI unità volumetrica (volumetric unit) | polygon | 25,497 | - |
| ISTAT Sassi area | polygon | 1 | - |
| ISTAT Census variables | polygon | 318 | - |
| Calcarenite Sampling Station | point | 8 | * |
| Down hole | point | 18 | * |
| HVNSR soil | point | 117 (10) | ** (*) [1] |
| HVNSR buildings | point | 96 (34) | ** (*) |
| MASW | point | 8 | * |
| Mechanical Surveys | point | 234 | * |
| Seismic Refraction Surveys | point | 7 | * |
| Surface features | point | 2 | - |
| Geomorphology | polygon | 301 | - |

**Table 1.** *Cont.*

| CLARA | Vector Geometry | # | Download |
|---|---|---|---|
| Geology | polygon | 13 | - |
| MOPS | polygon | 52 | - |
| Building resonance level | polygon | 4043 | - |
| Building frequency | polygon | 4043 | - |
| Soil isofrequency map | polygon | 7652 | - |
| Soil isoamplitude map | polygon | 7652 | - |
| DSM m (Orthometric Heights) | raster | - | - |
| DSM blg Height | raster | - | - |

[1] * and ** indicate pre-existing and new geological/geophysical downloadable data, respectively.

### 2.1. Open Data

Open data from the following three sources was used: (i) Regional Spatial Data Infrastructure of the Basilicata Region (RSDI) [55], (ii) OpenData (OD) Matera [56], and (iii) Italian National Institute of Statistics (ISTAT) [57]. RSDI is the main channel of the Basilicata Region for disseminating updated territorial information with technical and thematic cartographic production. OD Matera is a catalogue that allows users to search, access, download, and preview open data relative to the city of Matera through a single access point. ISTAT is a public research organization producing official statistics and operating in tandem with the academic and scientific communities. Two shapefiles from RSDI (original names: 'edifici_is' and 'unità volumetrica'), one from OD Matera portal (original name: rnc_4326.shp), and two from ISTAT (original names: 'R17_11_WGS84' and 'R17_indicatori_2011_sezioni') were downloaded and used for CLARA WebGIS. From the merging of the data contained therein, it was possible to obtain a new shapefile consisting of 4043 buildings, each with information relating to building typology (Figure 1A; masonry, reinforced concrete moment-resisting frame buildings, etc.), year of construction (Figure 1B), use and state of conservation (Figure 1C), and (eaves and maximum) heights (Figure 1D). ISTAT shapefiles contain a series of census variables, the municipal administrative limits, and the census sections of the study area, from which the SASSI area has been excluded (Figure 1).

For the 2648 buildings reported in OD Matera portal, there is a prevalence of reinforced concrete (~70%) compared to masonry (~27%) (Figure 1A). In general, a good (~51%) and very good (~42%) state of conservation is reported for residential buildings, with few buildings in poor and mediocre condition (~7%), which often corresponds to those of older construction (as of last release, 14 February 2018) (Figure 1B,C). The ISTAT data aggregated by neighbourhood allow us to make considerations and assessments on a territorial scale., i.e., the historic districts of the city (Piccianello, Historical Centre, Cappuccini-Agna) mainly consist of masonry dwellings, while the more recently urbanized districts show a prevalence of reinforced concrete moment-resisting frame buildings with a better state of conservation (Figure 2).

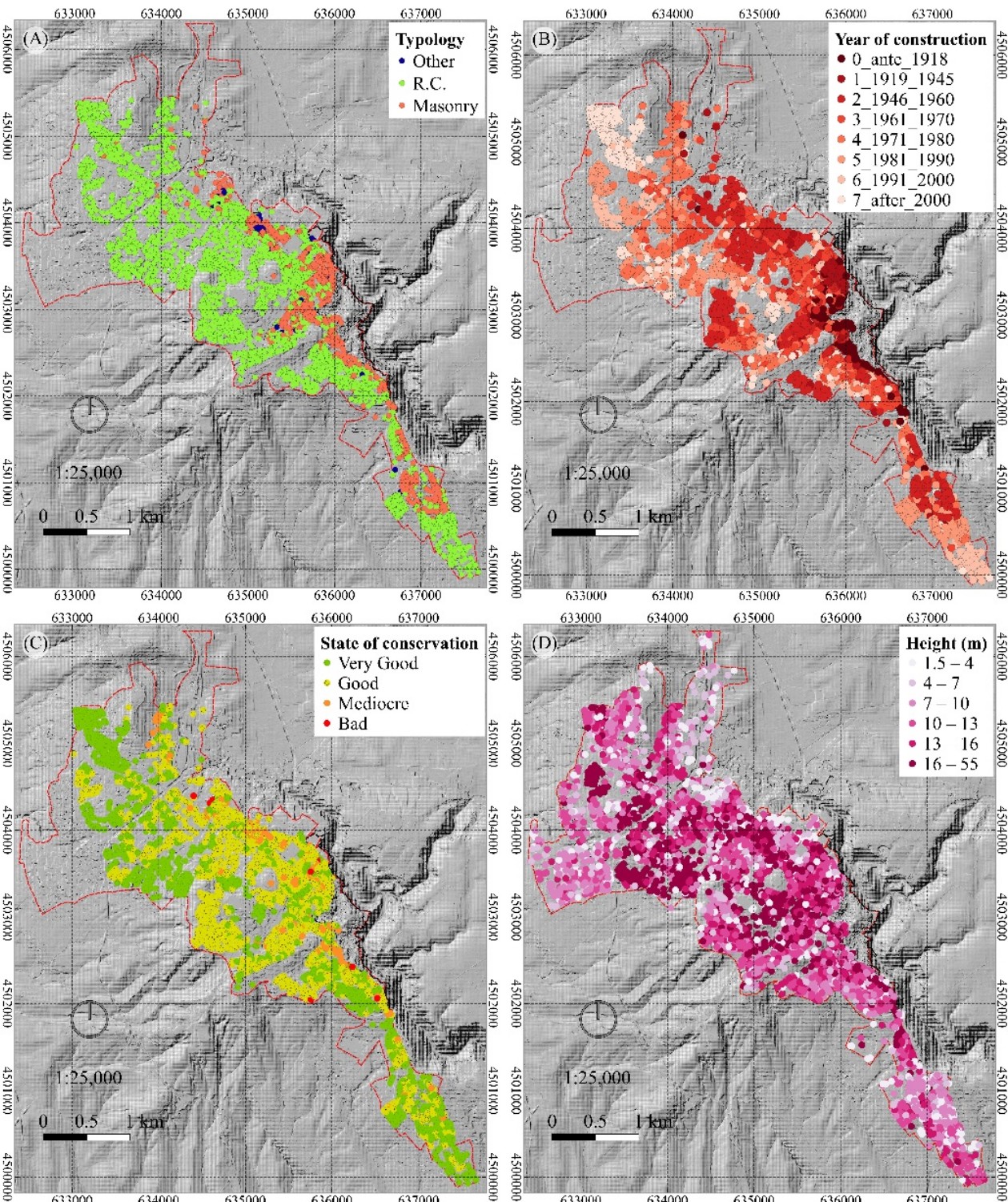

**Figure 1.** Spatial distribution of building characteristics: (**A**) building typology; (**B**) year of construction; (**C**) state of conservation; (**D**) height max (m). The black dashed line encloses the 'SASSI' area (not included in this study). The red dashed line delimits the urban area on which the study is focused.

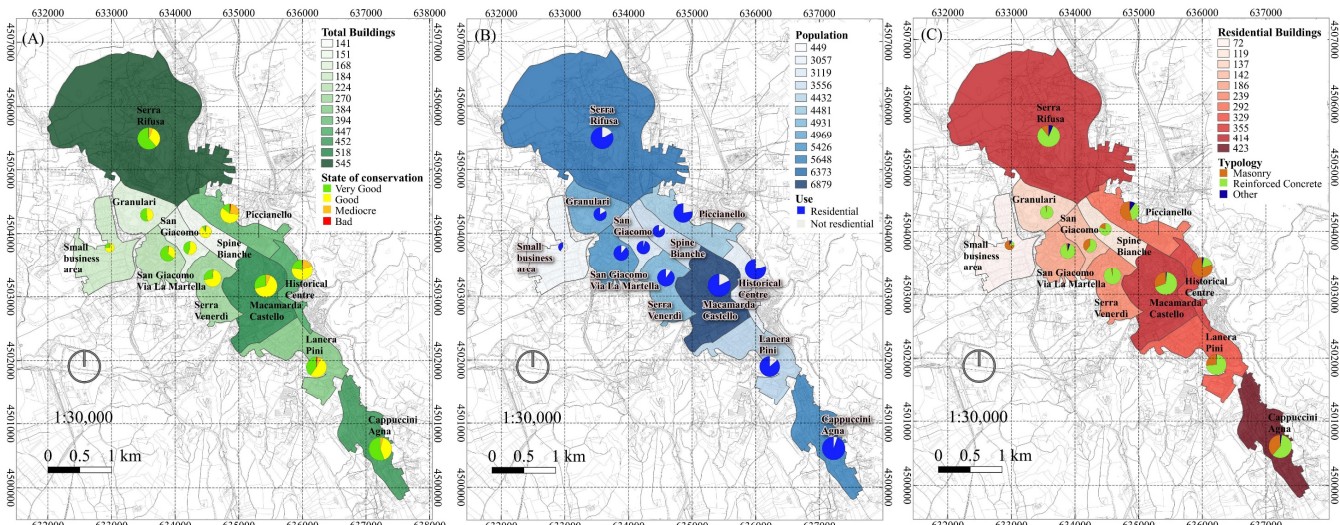

**Figure 2.** Spatial distribution of ISTAT census variables aggregated by neighborhood and classified according to (**A**) number of total buildings; (**B**) population; (**C**) number of residential buildings, including a pie chart of (**A**) building state of conservation, (**B**) building use, (**C**) built typology.

### 2.2. Pre-Existing Data

Examination, collection, digitization, and organization of the data available from previous studies were carried out for the area of Matera. An archive made up of 319 georeferenced geological, geotechnical, geophysical, and seismic surveys (downholes, mechanical surveys, calcarenite sampling stations, MASW, HVNSR, seismic refraction surveys; Figure 3) was harmonized in a geo-database; ~11% of which consisted of seismic surveys performed on buildings (Table 1). Moreover, the following maps of urban areas derived from microzonation studies were digitalized and georeferenced: geological, geomorphological, and MOPS (homogeneous microzones from a seismic response perspective). The user can access, visualize, query, and download data via the WebGIS user interface by clicking on the geometries. The information display mode is possible at all levels; after clicking on the geometry, the factsheets of all the active geometries arranged under the selected point will appear in nested mode (Figure 3). Data can be downloaded by clicking on the hyperlink contained in the last row of each attribute table. It is worth pointing out that ~87% of test certificates for geological-technical surveys are downloadable; for ~12.5%, only the main results are available.

### 2.3. CLARA Data

#### 2.3.1. Ground-Based Geophysical Data

In the framework of the CLARA project, the interaction effect between near-surface geology and all overlying buildings in the urban area of the city of Matera, as thoroughly described by [8], was evaluated. To this end, 107 single-station seismic ambient noise measurements of the main urban lithologies and 62 of the main building typologies were planned and performed. All the seismic ambient noise recordings on soils and buildings were performed using a Tromino seismograph (MoHo s.r.l.) and were analyzed applying the Horizontal-to-Vertical spectral ratio technique (HVNSR), following the standard procedures [58–60] and using Grilla software (version 8.1, Moho s.r.l. 2018). It was possible to estimate the main resonance frequencies for all urban soils and the main vibrational frequencies for the measured buildings.

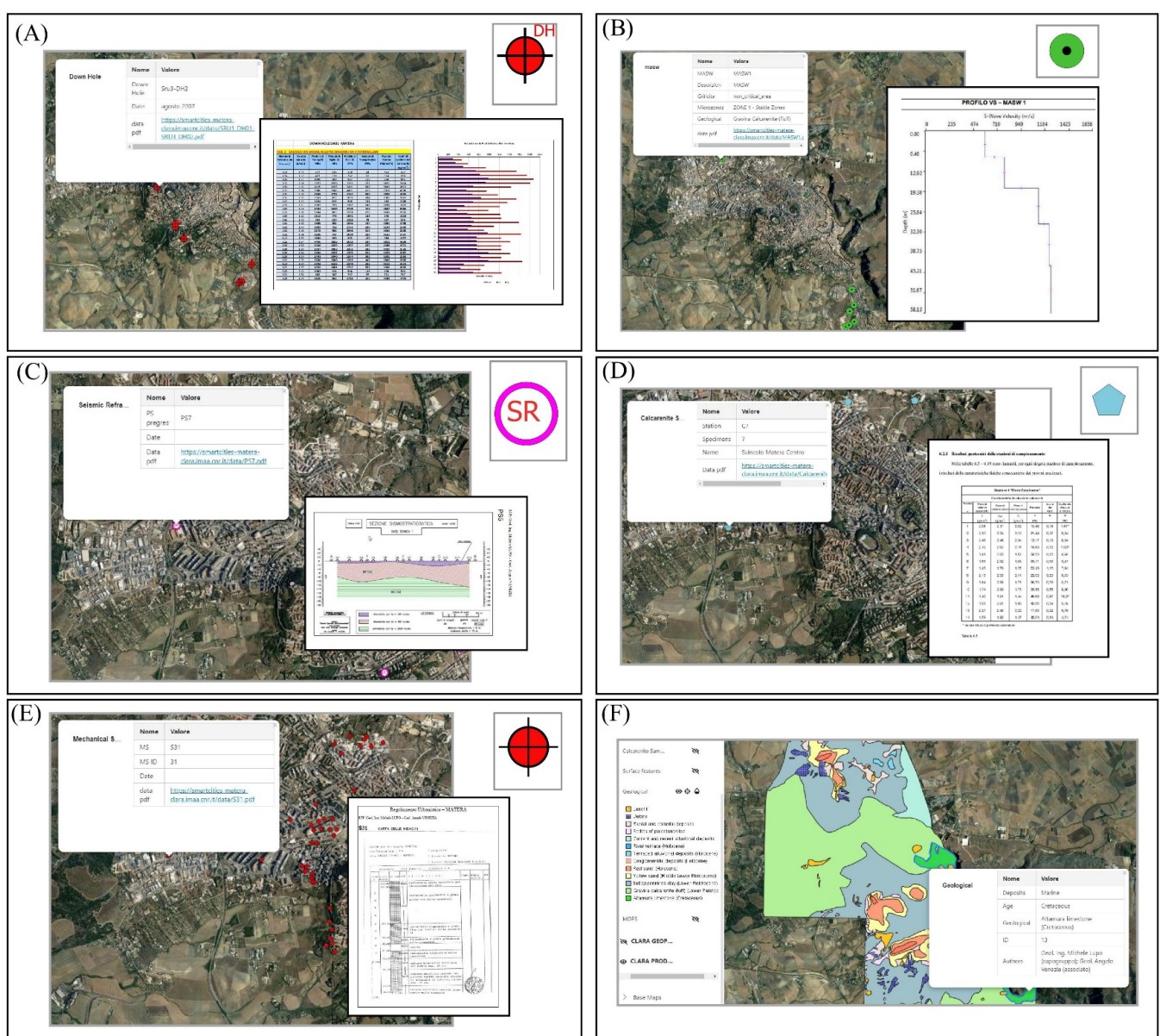

**Figure 3.** Examples of factsheets displayed by clicking on the related geometries (polygon in **F** sub-image) or symbols (upper right corner of single sub-images **A**–**E**) and examples of downloadable files of pre-existing geological/geotechnical/geophysical surveys for (**A**) downhole; (**B**) seismic refraction survey; (**C**) mechanical survey; (**D**) MASW; (**E**) calcarenite sampling station; (**F**) geological map.

Soil HVNSR

The integration of the 107 new and 10 pre-existing HVNSR (Table 1) curves evaluated for urban soils increased the area of the city of Matera covered by surveys from 18 to 1758 ha, with an average density of 3.7 surveys/km$^2$. Overall, the density of all surveys in the urban area is about 11.25 surveys/km$^2$. We merged in the same point-vector layer, named 'HVNSR soil', the pre-existing and new HVNSR functions. By clicking with the hand icon cursor on the point of interest, it is possible to visualize a factsheet with all the information related to that point (Figure 4). The thirteen fields shown in the factsheet have been given self-explanatory names: 'X' and 'Y' report the coordinates or the measurement locations in UTM WGS84 33N, EPSG 32633; 'A_Thresold' is the value of the HVNSR amplitude beyond which amplification is considered to occur (this value was chosen to be equal to 2 for all the analyses; Gallipoli et al., 2020 [8]); 'F0 soil' is the value of the soil

fundamental resonance frequency (Hz), with an amplitude equal to 'A0 soil'; 'F1 soil' (<'F0 soil') and 'F2 soil' (>'F0 soil') are the two frequencies (Hz) at which the HVNSR curve intersects the 'A_Thresold' value; 'Data_asc' and 'Data_bmp' contain hyperlinks to the downloadable HVNSR file in text format and HVNSR curve in bmp format, respectively; 'Macro area' indicates the neighbourhoods whose names are untranslatable, except for 'Historic Center' ('Centro Storico') and 'Small business area' ('Zona Artigianale') [61].

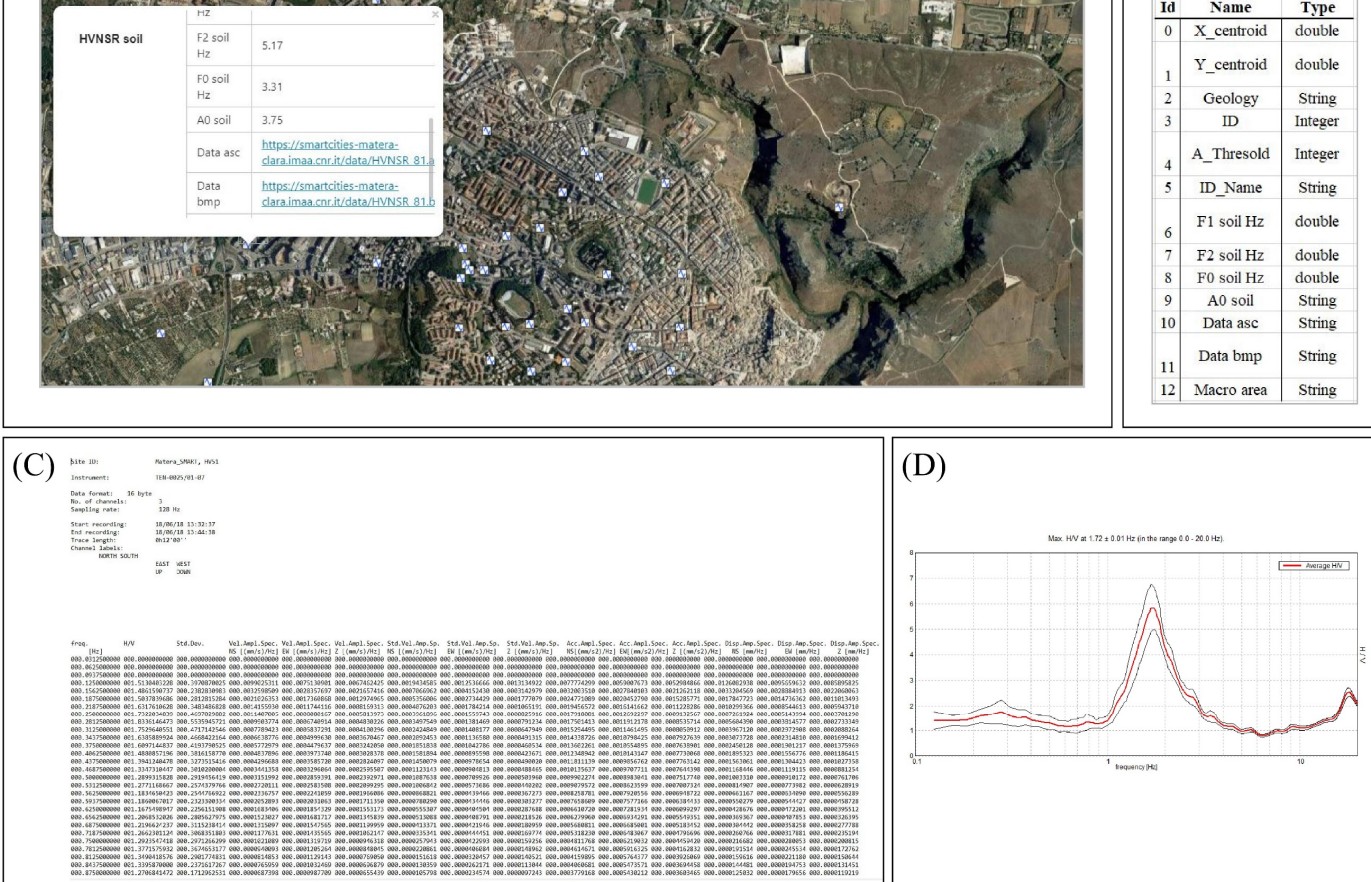

**Figure 4.** (**A**) Pop-up window showing the factsheet of a sample measurement (HVNSR_51) made on soil; (**B**) types of attribute; screenshots of downloadable (**C**) HVNSR text file and (**D**) HVNSR curve bmp file.

Building HVNSR

By integrating the 34 pre-existing with the 62 new HVNSR functions estimated for buildings, the percentage of buildings measured increased from 0.8% (#6) to 2.5% (#18) for masonry buildings (out of 732) and from 1.5% (#28) to 4.2% (#78) for reinforced concrete buildings (out of 1872). The percentage was calculated with respect to the total number of buildings falling within the studied area (#2648) for which the building typology was known. For reinforced concrete buildings, the sample distribution of measured buildings by macro-area is representative of the percentage of buildings in that area with respect to the total number of buildings in the urban study area (Figure 5B,C). The same does not apply for masonry buildings (Figure 5D,E) due to logistical issues (impossibility of access, etc.).

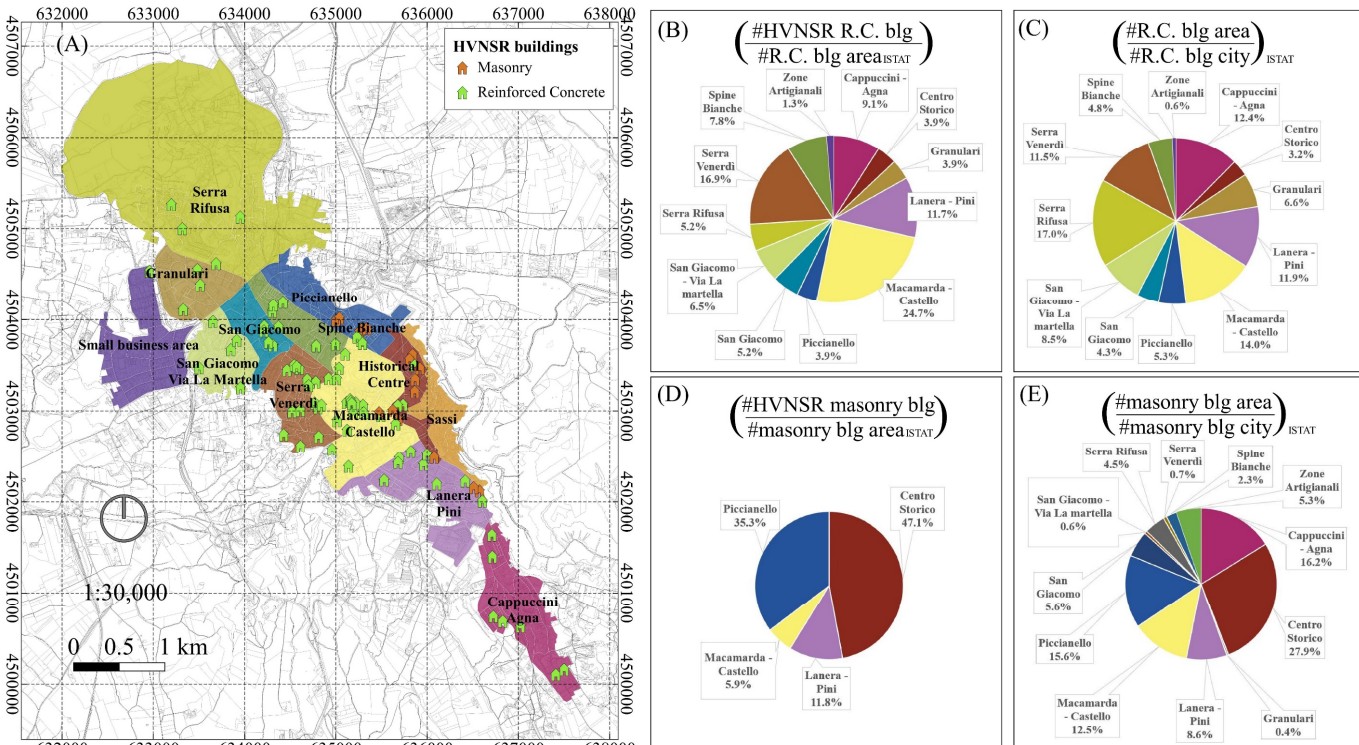

**Figure 5.** (**A**) Map of neighborhoods and measured building locations; pie charts showing proportion (as percentage) of measured and total number of buildings in the related macro-area for (**B**) reinforced concrete and (**D**) masonry; proportion (as percentage) of the number of buildings for each area with respect to the total number of buildings in the studied urban area for (**C**) reinforced concrete and (**E**) masonry, according to census variables (ISTAT).

Except for the buildings in the 'Granulari' districts or 'Small business area', which are almost completely founded on Gravina calcarenite, most of the buildings in the urban area of Matera lie on thick layers of Subappennines clays or above Gravina calcarenite (Figure 6). A detailed presentation of the lithostratigraphic features of the area is given by [8].

A total of 11 building HVNSRs in the same point-vector layer, named 'HVNSR buildings, were merged. The attribute table, available for consultation by clicking on a point, consist of fifteen fields with self-explanatory names (Figure 7): 'X' and 'Y' report the coordinates in UTM WGS84 33N, EPSG 32633; 'Type' refers to the building typology (R.C. or masonry); 'F0 blg Hz' is the main vibrational frequency of the building, retrieved by the HVNSR technique; 'Use' indicates the specific use for which a building is projected and built (i.e., residential, commercial, public, etc.); 'Data asc zip' and 'Data bmp' report the hyperlink of the downloadable HVNSR file in text format and HVNSR curve in bmp format, respectively.

### 2.3.2. Digital Surface Model from Satellite Data

In the framework of the CLARA project, we generated a DSM of the city of Matera and its surroundings, including the slope of the rocky ravine created by the Gravina stream [62]. In particular, the Agisoft Metahape photogrammetric software (Agisoft Metashape, 2021) [63] was used to process a cross-sensor multi-view satellite optical triplet composed of a WorldView-3 stereo pair and a GeoEye-1 image, whose acquisition features are described in [54].

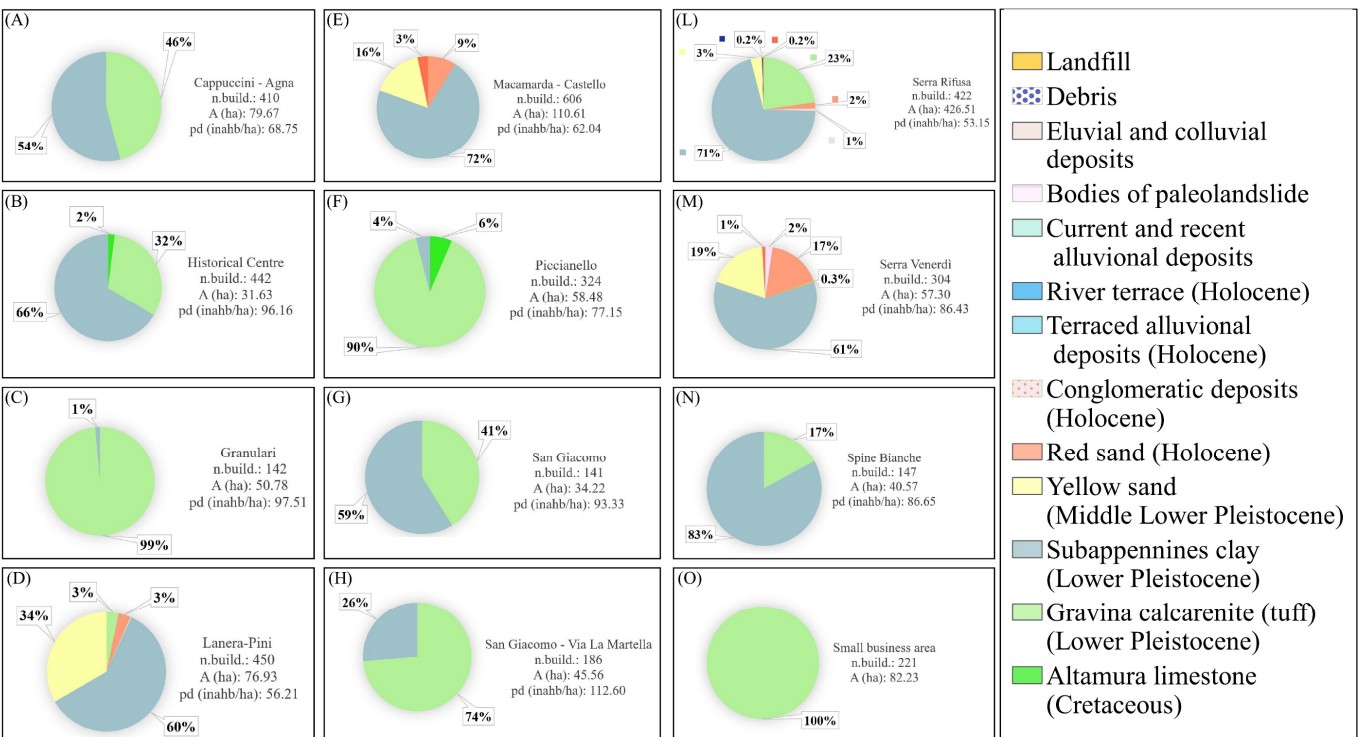

**Figure 6.** (**A**–**O**) Pie charts showing proportions (as percentages) of major foundation soil geology for each macro-area. Every diagram reports the macro-area name, the number of buildings (n.build.), the extension of the area in hectares (A(ha)), and the population density expressed as inhab/ha (pd).

First, three DSMs were produced using the three stereo pairs obtained by combining the three images. We followed the procedure described in [54], which adopts an original terrain-independent approach to refine the Rational Polynomial Coefficients (RPCs) supplied in each image metadata [64]. The geoid undulations derived from the EGM2008 model [65] were applied to transform the native ellipsoidal heights, derived from the RPC based orientation, to the corresponding orthometric ones. Moreover, we cropped the DSMs in accordance with the maximum intersection area common to the three DSMs, and we resampled our products to 0.5 m. Lastly, the final DSM was produced by computing a weighted mean of the three DSMs [54]. Moreover, an additional raster, containing the building heights with respect to the ground level for each pixel belonging to a building, was computed by subtracting the heights of the RSDI Digital Terrain Model (DTM)—resampled to the DSM resolution—from the DSM heights.

To evaluate the accuracy of the overall DSM, our product was compared with a reference generated using the open data from RSDI. Specifically, we added the orthometric heights of the RSDI DTM, resampled to the DSM resolution, to the building eave heights of the shapefile called 'unità volumetrica' to produce a reference DSM. In this way, the height differences ($\Delta Z$) between the reference and the overall DSM were computed for each pixel (Figure 8), and the standard statistical indicators [66] such as mean, standard deviation, root mean square error (RMSE), median, normalized median absolute deviation (NMAD), Linear Error at 68% and 90% confidence interval (LE68 and LE90, respectively) were evaluated. To compute the indicators, a threshold $\Delta Z$ equal to 20 m was adopted to remove the values outside the range ($-\Delta Z$, $\Delta Z$), which were considered outliers.

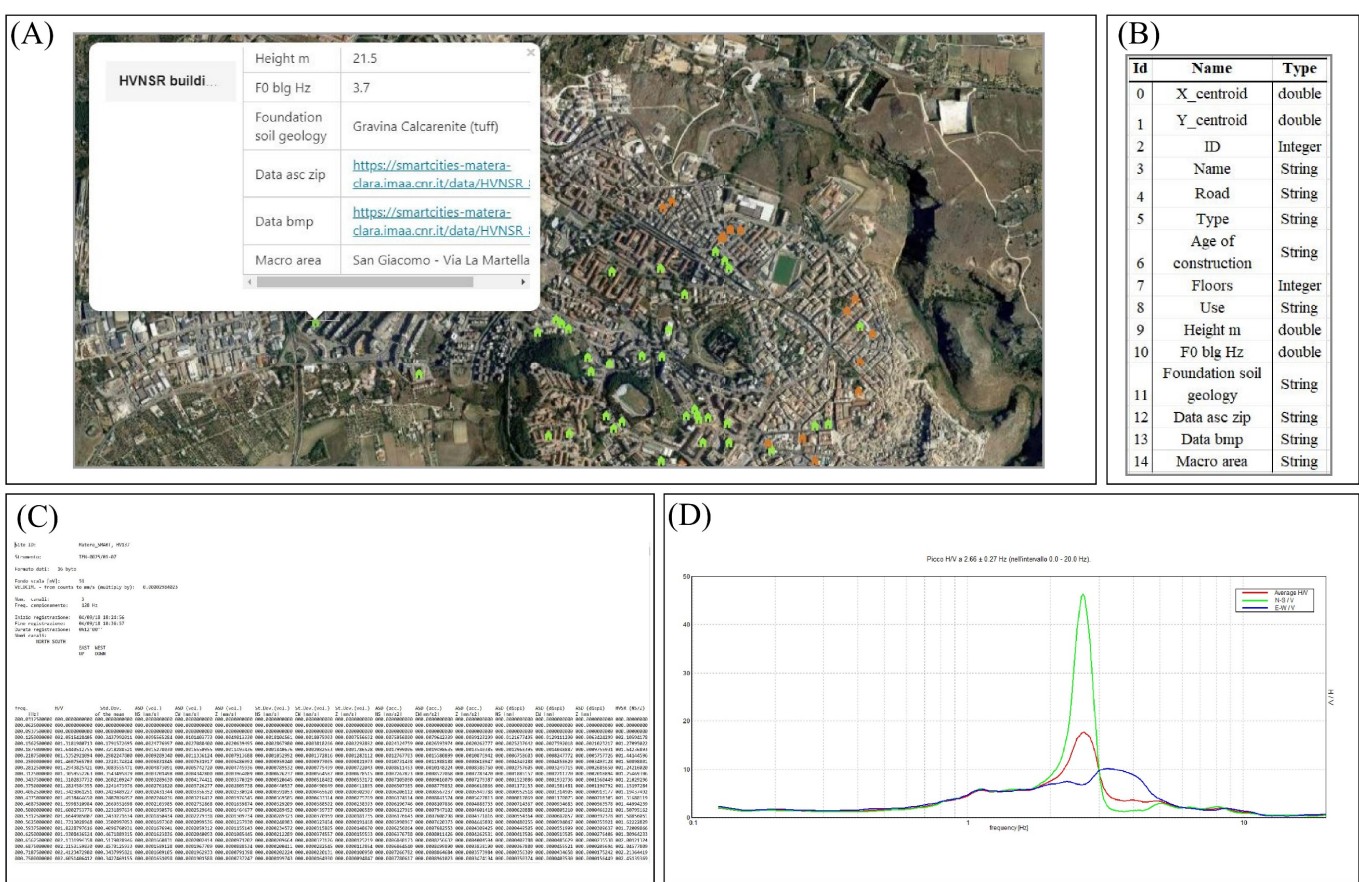

**Figure 7.** (**A**) Pop-up window showing the factsheet of a sample measurement (HVNSR_69) made on a R.C. building; (**B**) attribute types; screenshots of downloadable (**C**) HVNSR text file and (**D**) HVNSR curve bmp file.

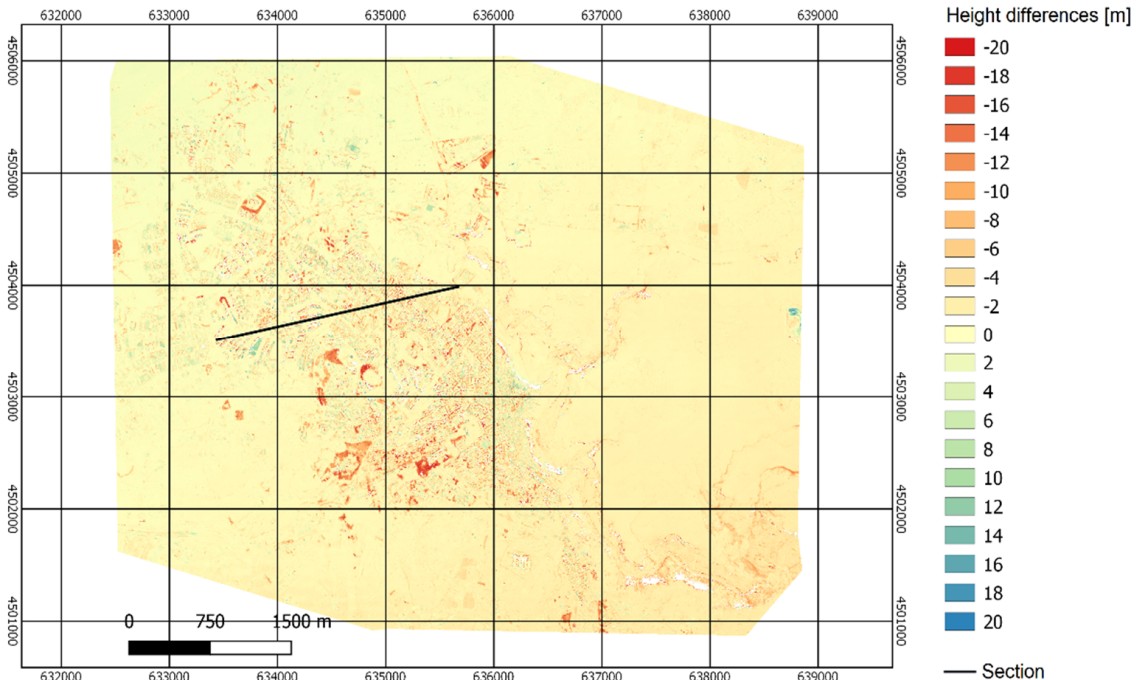

**Figure 8.** Error map (CRS: EPSG:32633-WGS 84/UTM, zone 33N-Projected) showing height differences (in m) between the reference and the obtained DSM. The black line represent a section detailed in Figure 10.

The results were assessed using the entire DSM and considering the different land covers separately. To distinguish artificial surfaces as well as agricultural, forest, and semi-natural areas, we adopted the CORINE Land Cover inventory [67] at epoch 2012, resampled to the DSM resolution. Furthermore, the statistics over the area corresponding exclusively to the buildings were computed using the 'unità volumetrica' shapefile. The results of the overall DSM validation are shown in Table 2. The DSM shows completeness equal to 98.42%, evaluated as the ratio (percentage) of the filled DSM pixels over the number of pixels of the reference raster.

**Table 2.** Statistical indicators for DSM accuracy assessment.

| Tile | Mean [m] | Std. Dev. [m] | RMSE [m] | Median [m] | NMAD [m] | LE68 [m] | LE90 [m] | Number of Pixels |
|---|---|---|---|---|---|---|---|---|
| Overall | −1.1 | 2.4 | 2.7 | −0.8 | 1.1 | 1.2 | 3.0 | 119,865,099 |
| Artificial surfaces | −1.3 | 3.7 | 3.9 | −0.6 | 1.5 | 2.0 | 5.9 | 34,110,194 |
| Agricultural areas | −0.8 | 1.6 | 1.8 | −0.8 | 0.9 | 0.9 | 1.8 | 68,037,393 |
| Semi-natural areas | −1.8 | 1.9 | 2.6 | −1.4 | 1.0 | 1.2 | 2.7 | 17,717,512 |
| Buildings | 1.2 | 3.4 | 3.7 | 0.8 | 2.3 | 2.6 | 5.4 | 8,110,798 |

It is worth noting that the errors in the semi-natural class can derive from the missing reconstruction of the vegetation in the reference DSM since it was generated from a DTM (Figure 9). Moreover, the 'Artificial Surfaces' and 'Buildings' classes are the most critical due to the high urban density, which makes the DSM production from satellite images very challenging because of occlusion issues (Figure 10). For these two classes, the RMSE reaches the highest values, being slightly higher than 3.5 m. However, in this case, the values of median and NMAD, which are significantly lower than the mean and standard deviation, denote the presence of outliers. As mentioned, most are probably related to occlusion issues; however, careful inspection showed that some outliers could be due to rather coarse simplifications inside the 'unità volumetrica' shapefile, which was used to generate the reference for the buildings. In fact, some complex buildings (one relevant example is the castle; Figure 9) were represented with just one unique height. Therefore, it is possible to improve the building heights in RSDI using the generated DSM where outliers were highlighted.

Finally, Figure 10 shows a section extracted from the reference (in black) and the generated DSM (in red); as indicated in the lower left and lower right corners, the profile direction is SW-NE (Figure 8). Even if the resulting DSM is noisier than the reference, the multi-view approach allows for the accurate and dense reconstruction of the terrain and all the objects within it (buildings—visible in the reference—and vegetation—not included).

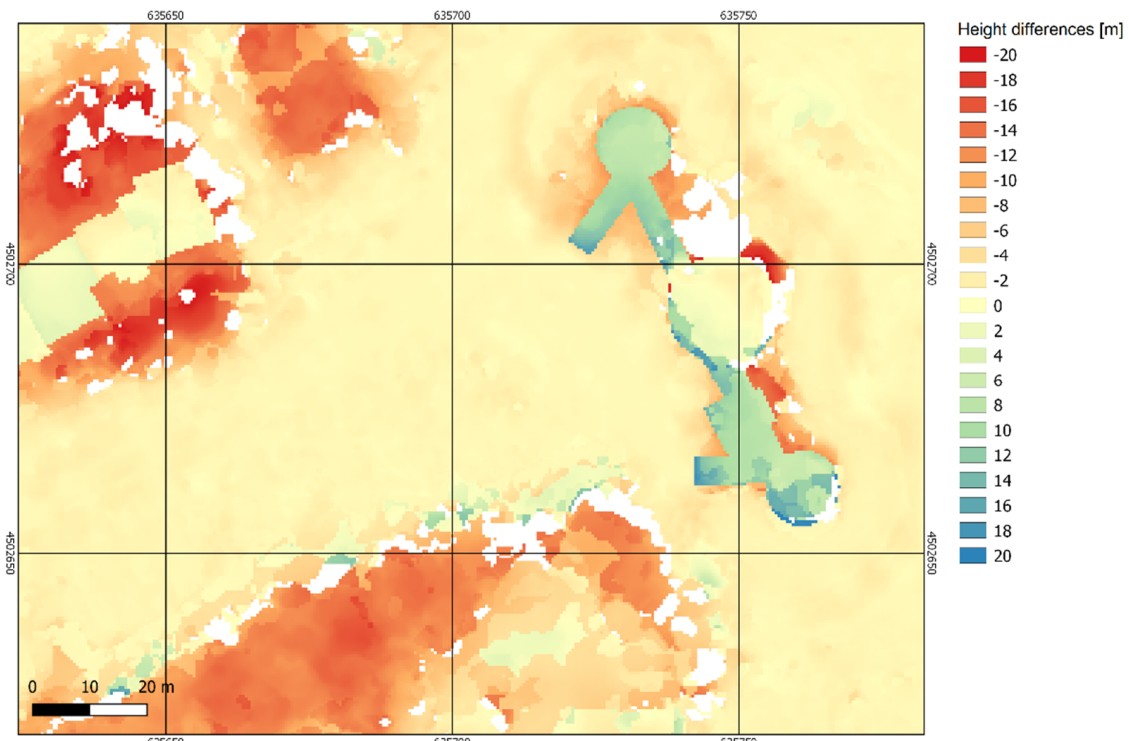

**Figure 9.** Error map (CRS: EPSG:32633-WGS 84/UTM, zone 33N-Projected) showing height differences (in m) between the reference and the obtained DSM: detail of the castle area, where the highest errors are due to the missing vegetation in the DTM and to the coarse unique height in the reference, derived from the 'unità volumetrica' shapefile (the two wings of the castle are shorter than the central tower, but they are registered with the same height in the shapefile).

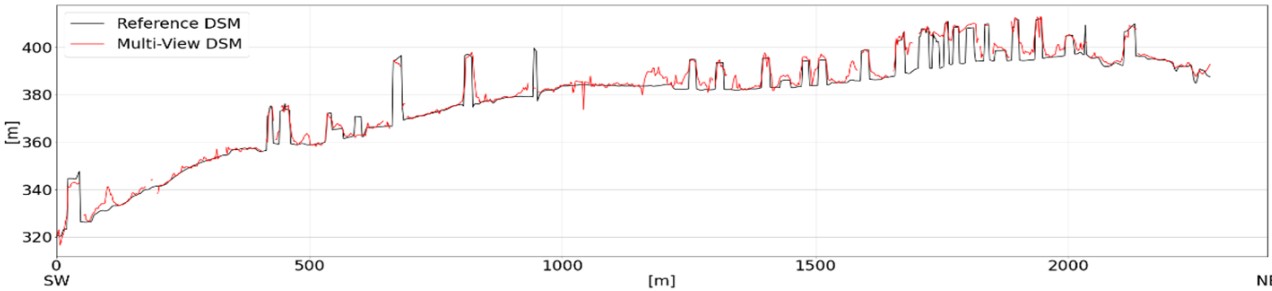

**Figure 10.** Section extracted from the reference DSM and the DSM generated with the multi-view approach.

## 3. CLARA WebGIS Products

### 3.1. Soil Isofrequency, Soil Iso-Amplitude, and Building Frequency Distribution Maps

Interpolating the soil resonance frequency values ('F0 soil Hz') and the relative amplitude values ('A0 soil'), which derive from the HVSR analysis of the measurements carried out at each of the 117 urban soil locations, the soil isofrequency (Figure 11A) and isoamplitude maps (Figure 11B), respectively, were retrieved. When clicking on any cell of the 'Soil isofrequency' (or 'Soil isoamplitude') map layer, a factsheet in multi-level form shows the interpolated 'F0 soil Hz' (or amplitude 'A0 soil') value. Using the empirical relationship T = 0.0167 H, estimated for the measured buildings [8] and having available the heights for all buildings of the Matera City, we predicted the main vibrational frequency and its uncertainty ('F0 blg range Hz') for 4043 buildings. By clicking on any geometry of the 'Building frequency' layer, the user can visualize both the estimated main vibrational frequency of the building 'F0 blg Hz' and the interpolated frequency of its foundation soil 'F0 soil Hz' values in the same factsheet (Figure 11C).

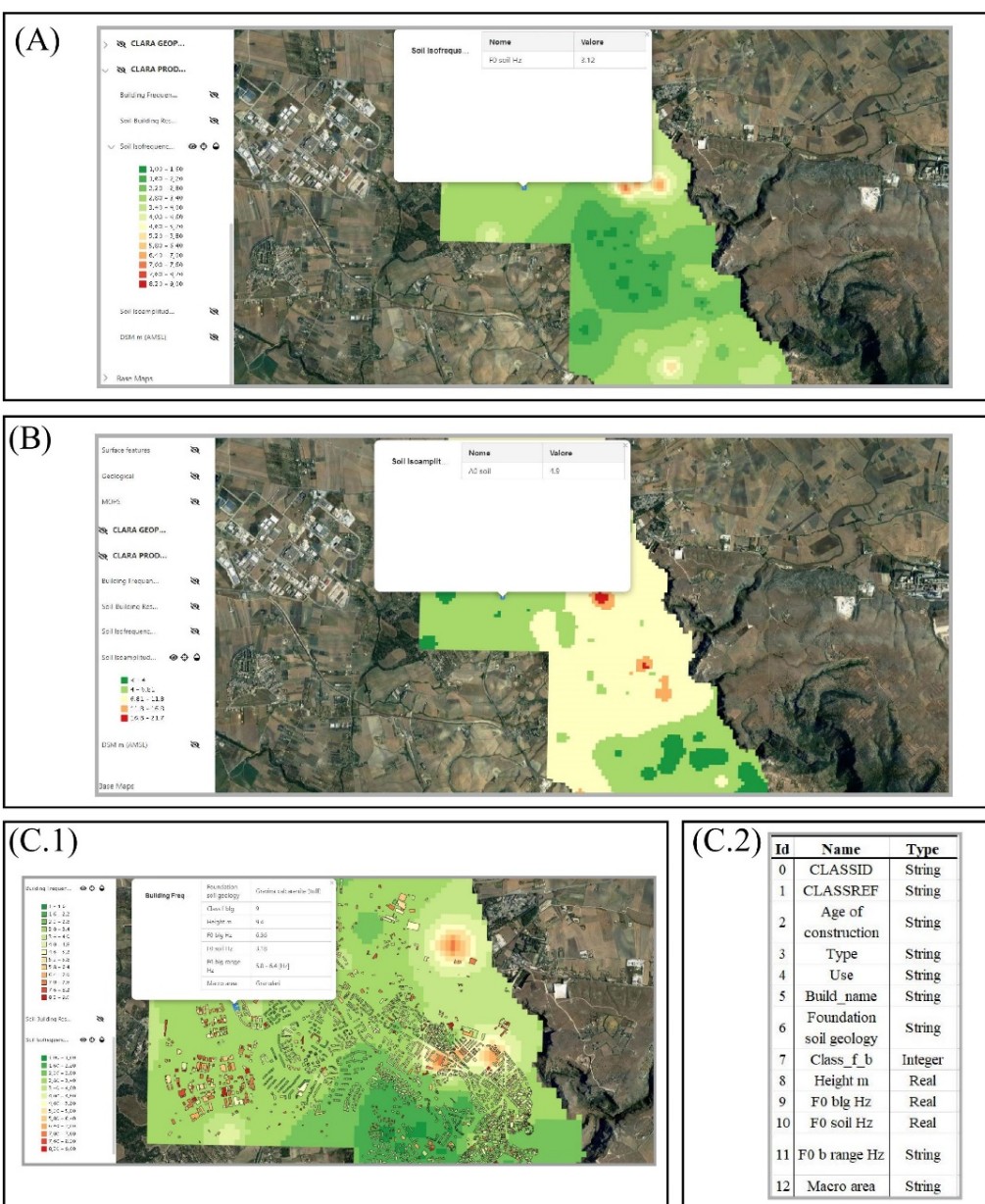

**Figure 11.** Screenshot of mapview and pop-up showing (**A**) soil isofrequency map; (**B**) soil isoamplitude map; (**C.1**) building frequency distribution overlying soil isofrequency map. The building frequency layer has an attribute table composed of thirteen fields (**C.2**).

The first vibrational frequencies of buildings ('F0 blg Hz) and the fundamental frequencies of soils ('F0 soil Hz') are classified based on the same frequency ranges and colour palette; when a building and the underlying soil pixel have the same color it means that they vibrate in the same range of frequencies (Figure 11C). This layer has an attribute table composed of thirteen fields (Figure 11(C.2)), with most of them inherited from the layers presented above. The new field, 'Class f b', indicates the class of frequency range ('F0 b range Hz') to which the building belongs, reported in the penultimate row of the factsheet.

## 3.2. Soil-Building Resonance Map

The soil-building resonance effect was evaluated considering the overlap between the amplifying HVNSR frequency ranges of the soil ('F1 soil Hz' and 'F2 soil Hz') and buildings ('F1 blg Hz' and 'F2 blg Hz'), as shown in [8]. Six levels of probability of soil-building resonance occurrence, encoded through a color scale ranging from green (low probability

of resonance occurrence) to red (high probability of resonance occurrence), were identified. A building whose estimated range of resonance falls within the interpolated resonance range of the underlying soil is attributed a high probability of resonance occurrence (100%, colored red), whereas a low probability level is assigned when the estimated resonance range of the building is completely disjointed from that of the interpolated underlying soil (Figure 12). When clicking on any of the 4043 polygons in the 'Soil–building resonance levels', an attribute table containing fourteen fields pops up. Most of the entries in the factsheet are inherited from other layers, except a new one, named 'resonance level', which reports the concatenation of three pieces of information: the class (of resonance occurrence probability) to which the building belongs, the total number of buildings in that same class, and the percentage with respect to the analyzed building stock (4043).

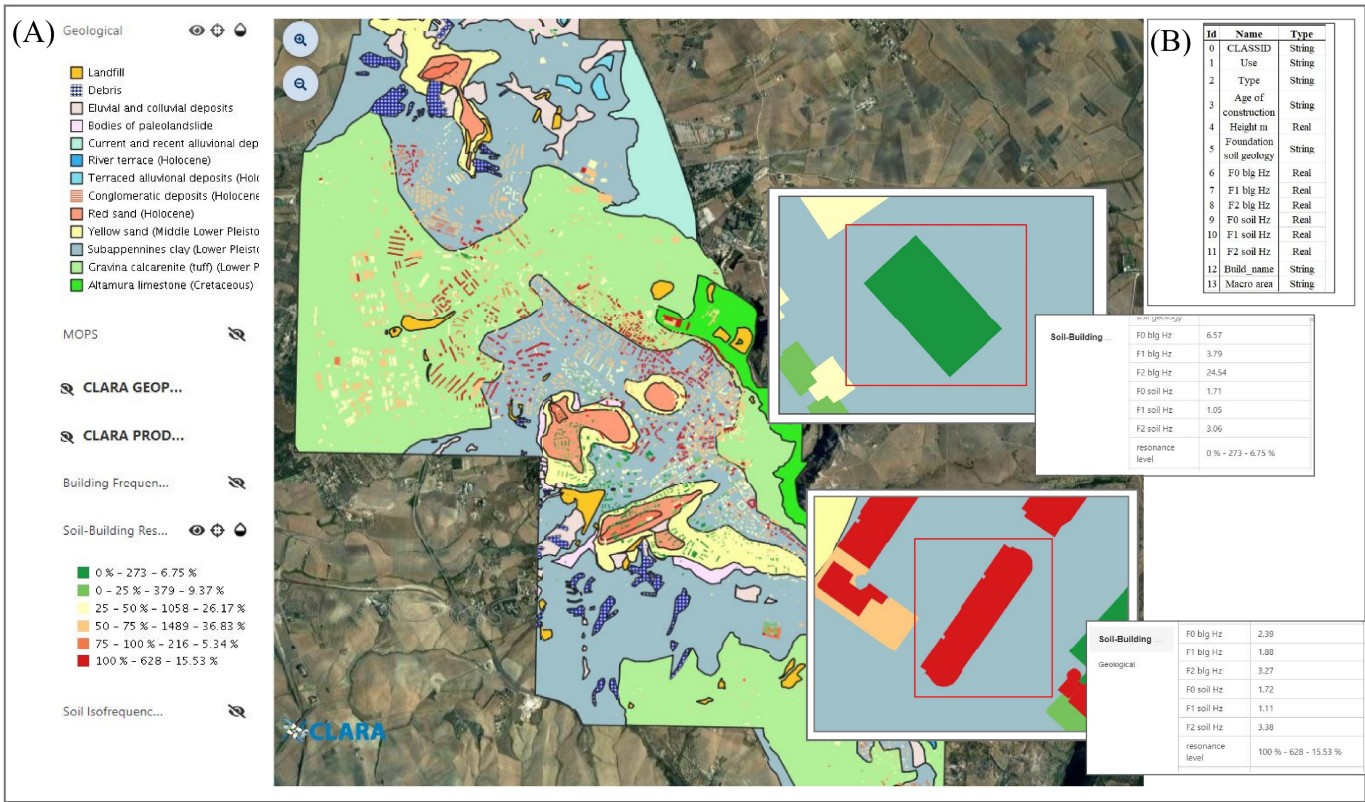

**Figure 12.** (**A**) Screenshot of soil–building resonance levels overlapping the geological map in the urban area of Matera. The insets show two examples: a (red-colored) building having 'F1 blg Hz' and 'F2 blg Hz' values included in the interpolated soil frequency range (100% probability of resonance occurrence), and a (green-colored) building with a probability of resonance effect equal to 0%, along with the related pop-up windows; (**B**) attribute table of soil-building resonance level layer.

### 3.3. Digital Surface Model and Building Height Rasters

The DSM raster has a resolution of 0.5 × 0.5 m and covers an area ranging from 632,450.2 m E to 638,868.7 m E and from 4,500,872.2 m N to 4,506,039.2 m N (Coordinate Reference System CRS: EPSG:32633−WGS 84/UTM, zone 33N−Projected). The elevations, expressed in meters, are orthometric heights referred to EGM2008 geoid model. Among the 121′879′530 pixels of the raster, 1′921′658 (1.58%) have no height information; these void pixels are indeed located within the urban area, near and among the buildings, and in the deepest valleys of the rocky ravine, where the occlusions and shadows have a negative impact on the photogrammetric matching process. By clicking on any point of the layer, it is possible to visualize the height value of each pixel inside the raster (Figure 13).

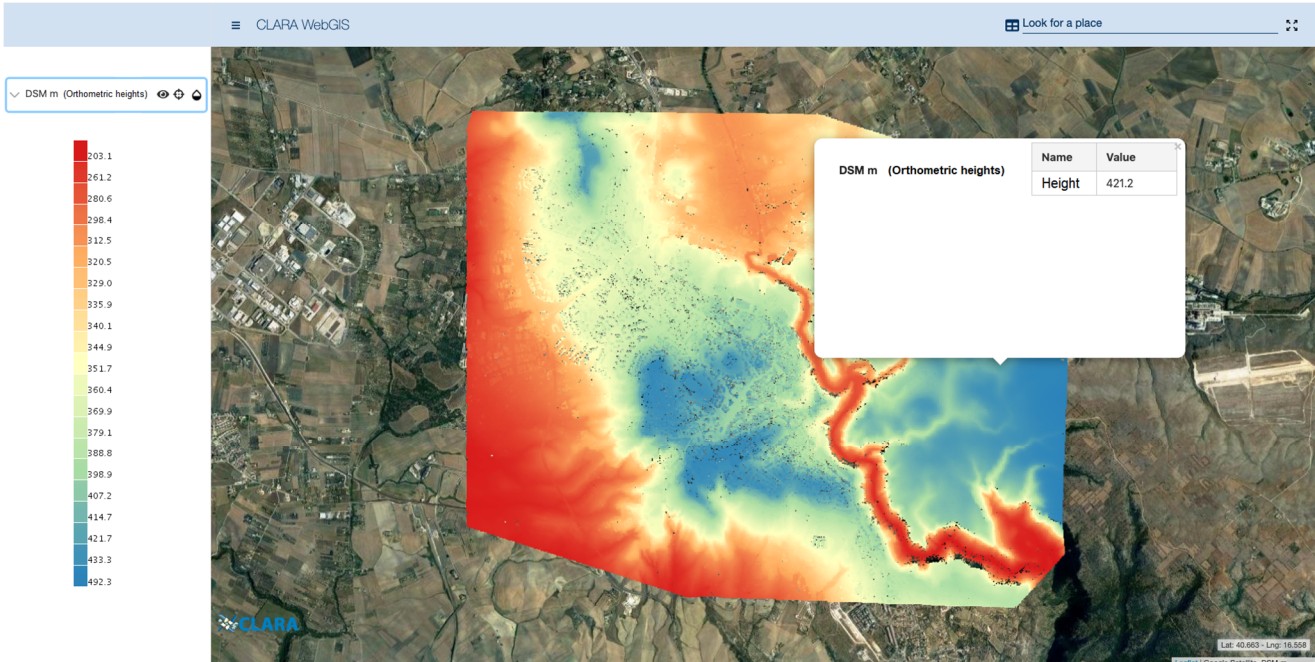

**Figure 13.** DSM raster (CRS: EPSG:32633−WGS 84/UTM, zone 33N−Projected).

The building height layer is a masked raster in which only the pixels corresponding to the buildings contain the height (expressed in meters) of the considered building with respect to the ground (Figure 14). It is important to highlight that this height is different from the eave height, which is the facade height (Figure 14). The building height raster covers the same area of the DSM raster and is characterized by the same resolution (0.5 m) and CRS.

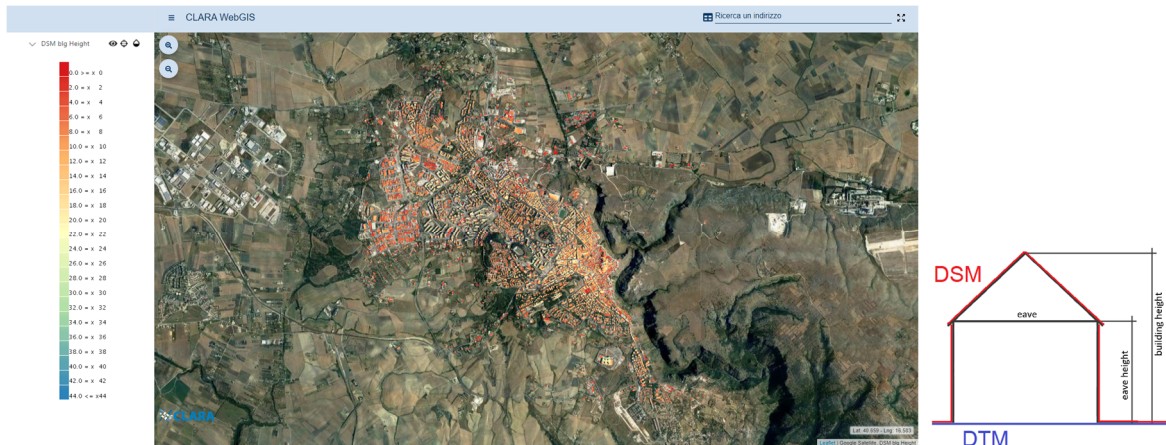

**Figure 14.** Building height raster (CRS: EPSG:32633−WGS 84/UTM, zone 33N−Projected). The building height is obtained as the difference between the DSM and the DTM and should not be confused with the eave height, i.e., the facade height.

## 4. Discussion

If societies do not learn from historical earthquakes and do not build up a culture of seismic risk management, earthquakes will continue to have catastrophic effects [68,69]. The improvement of the knowledge and awareness of individual citizens is key for achieving a better resilience of civil communities. In this view, sharing information about the seismic aspects specific to a given urban environment with the widest number of end users possible (central and local administrators and planners, engineers and professional geologists, citizens, etc.) is both vital and increasingly made possible by the emerging

paradigm of open data and modern geospatial technologies. Aimed at improving seismic risk mitigation in the city of Matera, CLARA WebGIS was designed to organize, analyze, and disseminate available information on soil, buildings, and soil–building interaction in the urban area of Matera by combining open data with new geospatial technologies. In the following section, we highlight some useful aspects identified in relation to those we believe to be the main users or beneficiaries of this tool, i.e., local authorities, urban planners, freelancers, and private citizens.

*CLARA WebGIS Potential and Perspectives*

The knowledge of the spatial distribution of the local seismic amplification effect, of the main characteristics of buildings, and of the soil-building resonance effect contributes efficiently achieving a three-part objective: (i) increasing the seismic resilience of an urban system and reducing the probability of a crisis occurring in the case of an earthquake; (ii) reducing the potential losses in economic and social terms; (iii) facilitating the return of the urban system to pre-existing conditions or recovering a new state of equilibrium by reducing the recovery phase time.

If we share the basic principle according to which the use of the areas affected by seismic amplification and their secondary effects (road obstruction, interruption of services, slowdown of rescue services, etc.) should be more carefully regulated, then the mapping of the probability of occurrence of soil-building resonance, along with the consequent assessment of the areas with the greatest probability of increased damage during seismic events becomes crucial for the implementation of mitigation and prevention strategies (urban planning laws, land-use planning, planning for intervention in emergencies and to manage post-earthquake crises). For example, the soil-structure interaction maps in the urban area of cities could help to (i) determine the most suitable areas for urbanization (characterized by low resonance levels) or eligible for other intended uses, such as parks, gardens, recreational areas (characterized by medium/low resonance); (ii) define seismic retrofitting strategies for existing strategic buildings/structures/infrastructures; (iii) integrate microzonation studies with the effects due to the presence of buildings and their interaction with the soil according to a holistic approach [8]. The achievement of the latter two targets is all the more feasible since some governments are currently financing seismic retrofitting for existing buildings and microzonation studies, e.g., the Italian government through the 'Sisma bonus' (Ministerial Decree August 6, 2020, n. 329) [70] and 'Guidelines for Seismic Microzonation' [71] respectively.

For years, technical and scientific communities have been discussing the opportunity of creating for each building a certificate containing all available information. Besides basic information on each building (height, age of construction, typology, use, etc.) and geological/geotechnical data of the relative foundation soil, the CLARA WebGIS contains the estimates of the fundamental frequency of all urban soils and the vibrational frequency in the linear elastic domain for 4043 buildings within the urban area. Such information constitutes invaluable knowledge for freelance engineers, as it is key for numerical models of seismic retrofitting.

In addition, CLARA WebGIS allows further evaluations in support of mitigation strategies, both on an urban and suburban scale, by combining the geophysical and engineering data contained in the layers. For example, by cross-referencing the data on the state of conservation of buildings with the probability levels of soil-building resonance occurrence, it is possible to estimate the number of buildings for which seismic retrofitting is recommended (Figure 15).

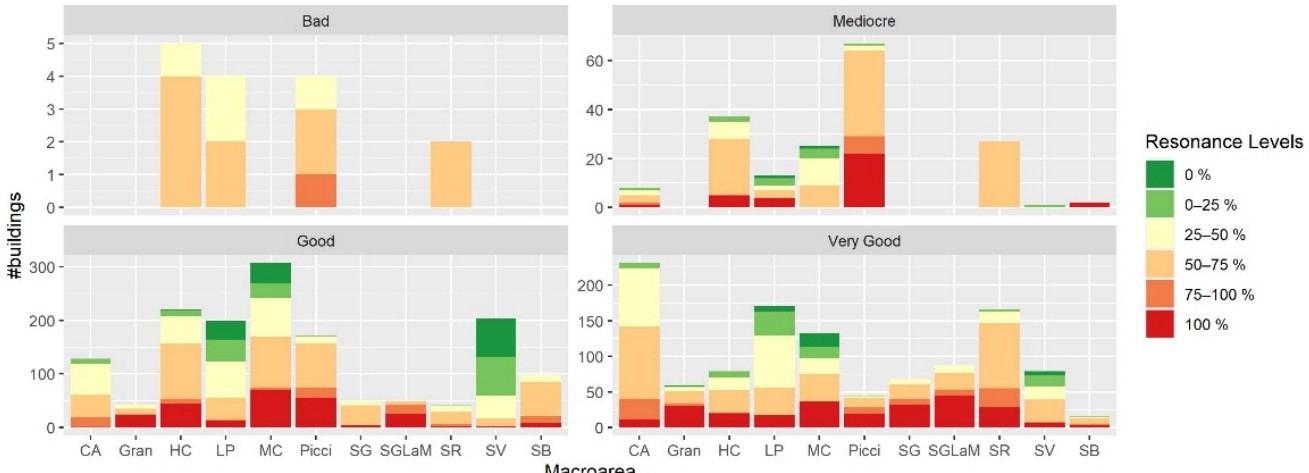

**Figure 15.** Bar charts showing the number of buildings for each macro-area, highlighting the probability levels of resonance occurrence and the building state of conservation. CA: Cappuccini-Agna; Gran: Granulari; HC: Historic Centre; LP: Lanera-Pini; MC: Macamarda-Castello; Picci: Piccianello; SG: San Giacomo; SGLaM: San Giacomo-Via La Martella; SR: Serra Rifusa; SV: Serra Venerdì; SB: Spine Bianche.

The experience gained in the development of the CLARA WebGIS with respect to the creation of a digital surface model (DSM) starting from the elaboration of satellite data, can constitute an adaptable reference for those situations in which there is no availability of open data on building heights. Indeed, if DSMs, which represent the Earth's surface with all the human-made objects on it (including precise information on the orthometric heights of buildings and infrastructure), are accompanied by the orthometric elevation of the topographic surface (provided by a digital terrain or elevation model—a DTM or a DEM), allow retrieving an accurate estimate of the height from ground for each building. Moreover, for DSMs generated from high-resolution satellite imagery, the RMSE of the height estimate can reach a few meters (up to 3.5 m, as is the case for this study, which corresponds to an overestimate or underestimate of one floor for each building). Thanks to this estimate (whose level of accuracy may be possibly increased in the coming years with the development of new sensors) and to detailed information on the type of foundation soil, it would still be possible to evaluate the resonance frequency of the buildings (inversely proportional to their heights) in areas where data on building heights are totally absent (or not publicly available) and thus to predict the type of response of each building to a seismic event.

We believe the following goals need to be pursued: (1) the CLARA WebGIS should be continuously developed and updated, taking into account additional needs, future challenges, user feedback, and the best available ICT; (2) the management of CLARA WebGIS should be entrusted to local administrators to ensure greater efficiency in its updating and maintenance and, above all, to strengthen awareness of the perception of risk in the actors responsible for implementing mitigation strategies.

## 5. Conclusions

This paper analyzed the interactive CLARA WebGIS, which is accessible at https://smartcities-matera-clara.imaa.cnr.it/, a useful tool developed, maintained, and enriched by CNR–IMAA, managed by CNR-GeoSDI, and built using open-source software and with a user-friendly interface addressed to a wide range of end users (government administrators and planners, engineers and geologists, citizens, etc.). CLARA WebGIS lets users query and download 319 geological and geotechnical surveys (Downholes, Mechanical Surveys, Calcarenite Sampling Stations, MASW, HVNSR, Seismic Refraction Surveys) from studies conducted from 1990 to 2010, 213 new single-station seismic ambient noise measurements carried out between 2015 and 2019 during the CLARA project, geological and geomorpho-

logical maps, and a map of homogeneous microzones from a seismic perspective. The principal outputs derived by crossing all geophysical and engineering data available in the database are:

- the estimation of fundamental resonance frequencies for all urban soils;
- the estimation of the main vibrational frequencies for 4043 overlying buildings;
- the resonance effect of each building with respect to the relative foundation soil;
- the DSM generated using satellite imagery composed of a WorldView-3 stereo pair, a GeoEye-1 image, and a building height map obtained from the produced DSM and RSDI open data.

The first three outputs, regarding the main soil and building characteristics and their interaction, represent a key element to plan strategies for seismic risk mitigation in terms of urban planning, seismic retrofitting, and management of post-earthquake crises. Moreover, the detailed DSM could represent improved knowledge for those cities/megacities without open data on building heights. We hope that this tool can be a starting point for the administration of all cities and that individual geodatabases similar to CLARA WebGIS can be built by combining pre-existing and new geophysical data for the characterization of the soil and buildings.

**Author Contributions:** Conceptualization, M.R.G.; methodology, G.C., L.L., V.B., R.R. and M.R.G.; software, N.T., G.C., L.L., V.B., R.R. and V.S.; validation, L.L., V.B., R.R. and M.R.G.; formal analysis, N.T., G.C., L.L., V.B. and R.R.; investigation, N.T., G.C. and M.L.; resources, N.T., G.C., M.L., V.S. and M.R.G.; data curation, N.T., G.C., L.L., V.B. and R.R; writing—original draft preparation, N.T., G.C., L.L., V.B., R.R. and M.R.G.; writing—review and editing, G.C. and M.R.G.; visualization, N.T., G.C., L.L., V.B. and R.R.; supervision, M.R.G.; project administration, M.R.G.; funding acquisition, M.R.G. All authors have read and agreed to the published version of the manuscript.

**Funding:** This research was funded by project CLARA (Cloud pLAtform and smart underground imaging for natural Risk Assessment, id code n. SCN_00451, funded by the Italian Ministry of University and Research).

**Institutional Review Board Statement:** Not Applicable.

**Informed Consent Statement:** Not Applicable.

**Data Availability Statement:** The data presented in this study are openly available in https://smartcities-matera-clara.imaa.cnr.it/.

**Acknowledgments:** We would like to thank our research colleagues Angela Perrone (CNR IMAA, Tito Scalo-Potenza) and Mattia Crespi (Geodesy and Geomatics Division, DICEA, Sapienza University of Rome, Rome, Italy) for their helpful suggestions to improve the manuscript. The authors gratefully acknowledge the geoSDI team, who edited the online publication of WebGIS, particularly Donato Maio and Eng. Lorenzo Amato for their valuable support, cooperation, and precious assistance in the activities of the platform. A final word of gratitude is expressed to the owners of the buildings, who kindly provided their valuable time to let us do ambient noise measurements.

**Conflicts of Interest:** The authors declare no conflict of interest.

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
