# Peer review of "Sharing Soil and Building Geophysical Data for Seismic Characterization of Cities Using CLARA WebGIS: A Case Study of Matera (Southern Italy)"

_applsci, doi:10.3390/app11094254_

Round 1

Reviewer 1 Report

This WebGIS is an important contribution to the society to be prepared for seismic risk. The introduction of WebGIS is well written in the manuscript.  

Reviewer 2 Report

I recommend improving the manuscript taking into account the following comments and suggestions:

1. Title: Sharing soil and building geophysical data for seismic characterization of the city using CLARA WebGIS: a case study of Matera (Southern Italy)

2. Please do not use "We" in the whole paper. Please use the passive voice. 

3. The introduction should be shortened, for example: 
- text in lines 44-49 can be omitted. 
- text in lines 55-58 can be omitted (there is repeated in founding information). 
- description of the CLARA project (lines 59-72) should be shortened. 

4. Last paragraph (starting from line 109): Please indicate what is new in the CLARA project vs. other similar projects using Web GIS?

5. Section 2 should be reorganized. Some information is not necessary and causes confusion. 

6. Table 1: What do signs *, **, (*) in the last column mean? It should be explained below the table.

7. Figures 2, 3, 4, 5, 6, 7, 11, 12: Please add the better quality of all graphs because the text and tables inserted into the figures are illegible.

8. Conclusions require reorganization. The text in lines 522–535 is repeated from the abstract and introduction. Please provide specific conclusions from the study (in points to make them understandable).
